# Electroosmotic Flow Behavior of Viscoelastic LPTT Fluid in a Microchannel

**DOI:** 10.3390/mi10120881

**Published:** 2019-12-15

**Authors:** Dilin Chen, Jie Li, Haiwen Chen, Lai Zhang, Hongna Zhang, Yu Ma

**Affiliations:** 1School of Energy and Power Engineering, Wuhan University of Technology, Wuhan 430070, China; 132979767518@163.com (D.C.); chw168920@163.com (H.C.); lai_zhang_123@163.com (L.Z.); 2Institut Franco-Chinois de l’Energie Nucléaire, Sun Yat-sen University, Zhuhai 519000, China

**Keywords:** electroosmotic flow, viscoelastic fluid, Linear Phan-Thien–Tanner (LPTT), pH, electrical double layer

## Abstract

In many research works, the fluid medium in electroosmosis is considered to be a Newtonian fluid, while the polymer solutions and biological fluids used in biomedical fields mostly belong to the non-Newtonian category. Based on the finite volume method (FVM), the electroosmotic flow (EOF) of viscoelastic fluids in near-neutral (pH = 7.5) solution considering four ions (K^+^, Cl^−^, H^+^, OH^−^) is numerically studied, as well as the viscoelastic fluids’ flow characteristics in a microchannel described by the Linear Phan-Thien–Tanner (LPTT) constitutive model under different conditions, including the electrical double layer (EDL) thickness, the Weissenberg number (*Wi*), the viscosity ratio and the polymer extensibility parameters. When the EDL does not overlap, the velocity profiles for both Newtonian and viscoelastic fluids are plug-like and increase sharply near the charged wall. Compared with Newtonian fluid at *Wi* = 3, the viscoelastic fluid velocity increases by 5 times and 9 times, respectively, under the EDL conditions of *kH* = 15 and *kH* = 250, indicating the shear thinning behavior of LPTT fluid. Shear stress obviously depends on the viscosity ratio and different *Wi* number conditions. The EOF is also enhanced by the increase (decrease) in polymer extensibility parameters (viscosity ratio). When the extensibility parameters are large, the contribution to velocity is gradually weakened.

## 1. Introduction

In recent years, with the development of multi-disciplines and mutual integration, microfluidic chips have become more widespread. In the shipping industry, broad application prospects can be foreseen in the Point-of-Care Testing (PoCT) of seafarers’ diseases [1,2] and the desalination [3,4] of automatic signal transmission. In addition, microfluidic chips have many applications in the sensing and detection of nanoparticles [5,6,7], operations and reactions in analytical chemistry [8,9], encryption and decryption of information, future military communication, etc. The electrodynamic transport characteristics of micro- and nanofluidic devices have important basic and practical significance. Reuss [10] first discovered the electroosmotic flow (EOF) phenomenon in the laboratory. Helmholtz [11] considered the liquid and electric flow phenomena comprehensively for the first time, connecting the electric phenomenon, fluid flow and ion concentration, and proposing a complete electrical double layer (EDL) model theory. When direct current (DC) or alternating current (*AC*) voltage is applied along the horizontal direction of the microchannel, the direction of the external electric field is tangent to the charged surface, and the liquid ions in the EDL will be directionally displaced by the electric field force, thus driving the fluid microclusters to move along the direction of the electric field force to form EOF.

Medical diagnostics include testing biological fluids such as saliva, blood and mucus, which exhibit viscoelastic behavior [12,13,14]. A growing number of PoCT methods involve the study of biological samples on laboratory microfluidic devices [15,16]. From the view of the chip-based microfluidic device laboratory, electroosmotic flow activity has been well studied and analyzed [17,18]. The use of simplified Possion–Boltzmann (PB) and Debye–Hückel (DH) models [19,20,21,22] for the analysis of EOF has made great progress, and the different solutions are extensive [23]. Yossifon et al. [24] analyzed the nonlinear current-voltage characteristics of nanochannels. Ma et al. [25] and Yeh et al. [26] proposed an analytical solution for the EOF of Newtonian fluids with multiple ion and pH characteristics. Huang et al. [27] first deduced a model to study the zeta potential and ionic conductivity of cylindrical nanopores with overlapping EDL. However, most biological fluids exhibit viscoelastic behavior [12,13,14]. Their non-Newtonian fluid characteristics are quite different from Newtonian fluids, including their shear rate-dependent viscosity, memory effect, and principal stress difference [28,29]. Therefore, the EOF of this kind of fluid should be distinguished from the traditional Newtonian fluid. Recently, more EOF theoretical studies on micro/nanofluidics have considered non-Newtonian properties. Das et al. [30] used the power law model to derive the analytical solution of the EOF in a rectangular microchannel. Zimmerman et al. [31] numerically simulated the electric flow behavior in a microchannel with a T-junction using the Carreau–Yasuda model. Berli and Olivares [32] used a multi-parameter non-Newtonian model to analyze the EOF in straight and cylindrical pipes. Zhao et al. [33] derived the analytical solution of power-law fluids’ EOF and analyzed the effects of generic flow behavior index (*n*). Li et al. [34] studied the transient flow of Maxwell fluid in a long, straight microchannel and the effect of relaxation time. Mei et al. [35] analyzed the EOF characteristics of Linear Phan-Thien–Tanner (LPTT) fluids in a nanoslit with EDL overlap.

Most of the previous viscoelastic fluid EOF studies neglected the introduction of pH and H^+^ and OH^−^ ions [17,18,19,20,21,22,23,24,36,37,38,39,40,41], or made an inaccurate characterization of biological fluid parameters [25,26,27] by a Newtonian fluid constitutive model (under pH conditions). Nowadays, a large number of biological experimental operations and chemical detection processes consider near-neutral conditions (pH = 7.5), such as specific drug release [42,43,44], biological fluid separation operation and amplification detection [45,46], sensor performance improvement [47,48,49,50], maintaining good activity of a microbial solution [51], and so on. Therefore, studying the EOF of viscoelastic fluid under near-neutral pH conditions may be far-reaching. Based on the finite volume method (FVM), the non-simplified Poisson–Nernst–Planck (PNP) model is used to describe the ion transport behavior. The LPTT constitutive model is considered to describe the viscoelastic properties, and a solution of pH 7.5 with H^+^ and OH^−^ ions is mainly considered. The Weissenberg number (*Wi*), EDL thickness, viscosity ratio, extensibility parameters and net charge density are numerically studied. The mathematical model and boundary conditions are given in Section 2, and the numerical method is verified in Section 3. Finally, the parameter research and conclusions are presented.

## 2. Mathematical Model and Boundary Conditions

The motion of incompressible viscoelastic fluid with near-neutral solution (pH = 7.5) is considered in a long channel with length *L*, height *H* and width *W*, including K^+^, Cl^−^, H^+^ and OH^-^ ions under the applied potential V_0_ across the channel. Assuming that the channel height is much less than the length and the width (*H* << *L*, *H* << *W*), the problem can be simplified to a 2D problem, shown in Figure 1. Cartesian coordinates are used, the y-axis is the direction of channel height *H*, the x-axis is the direction of length, and the origin is fixed on the symmetrical line (y = *H*).

The continuity equation and the momentum equation are as follows:(1)∇⋅u=0
(2)$$-\nabla p+2\eta_{s}\nabla \cdot D+\nabla \cdot \tau +\rho_{e}E=0$$

In the above, u and p are velocity and pressure, respectively; $\eta_{s}$ is the dynamic viscosity of the solvent; $D=\frac{1}{2}\left[\nabla u+{\left(\nabla u\right)}^{T}\right]$ is a deformation tensor; $\rho_{e}$ is the charge density of the electrolyte solution; E=−∇ψ represents electric field, where ψ is the potential in solution; τ is an extra stress tensor and can be described by different constitutive models, such as Oldryod-B, FENE-P and PPT. τ can usually be written as a function of the conformation tensor ***c***, and ***c*** is the tensor variable of the macromolecular structure of the polymer; the LPPT model is used to describe the viscoelastic fluid in this section.
(3)$$\tau =\frac{\eta_{p}}{\lambda }\left(c-I\right)$$
where $\eta_{p}$ is the polymer stress tensor and λ is the relaxation time.

The conformation tensor (***c***) for the LPTT model is governed by
(4)$$u\nabla \cdot c-\left(c\cdot \nabla u^{{}_{T}}+\nabla u\cdot c\right)=-\frac{1}{\lambda }\left(1+\varepsilon \left(tr\left(c\right)-3\right)\right)\left(c-I\right)$$
where ε is the extensibility parameter and tr(c) is the trace of the conformation tensor ***c***.

The potential ψ in the electrolyte solution is controlled by the Poisson equation:(5)$$-\varepsilon_{f}\nabla^{2}\psi =F\sum_{i=1}^{i=4}z_{i}c_{i}$$
where $\varepsilon_{f}$ is fluid permittivity; F is the Faraday constant; $c_{i}$ and $z_{i}$ (*i* = 1, 2, 3, 4) are the concentration and charge valence, respectively. The transport of ion concentrations is controlled by the Nernst–Planck equation:(6)$$\nabla \cdot N_{j}=\nabla \cdot (uc_{i}-D_{i}\nabla c_{i}-z_{i}\frac{D_{i}}{RT}Fc_{i}\nabla \psi )=0$$
where *R* is the general gas constant; *T* is the absolute temperature; *Di* is the diffuse rate of the i-th ion, and $N_{j}=uc_{i}-D_{i}\nabla c_{i}-z_{i}\frac{D_{i}}{RT}Fc_{i}\nabla \psi ,(j=1,...4)$ indicates each ion flux.

The KCl solution bulk concentration $C_{0}$ was selected as the ion concentration scale, *RT/F* as the potential scale, channel height *H* as the length scale, $u_{0}=\varepsilon_{f}R^{2}T^{2}/(2\eta_{0}HF^{2})$ as the velocity scale, $\eta_{0}=\eta_{s}+\eta_{p}$ as zero-shear rate total viscosity, and $\rho u_{0}{}^{2}$ as the pressure scale. The normalized governing equations (1), (2) and (4)–(6) can be written:(7)$$\nabla^{\prime }\cdot u^{\prime }=0$$
(8)$$u^{\prime }\cdot \nabla^{\prime }u^{\prime }=-\nabla^{\prime }p^{\prime }+\frac{\beta }{Re}{\nabla^{\prime }}^{2}u^{\prime }+\frac{\left(1-\beta \right)}{Re\cdot Wi}\nabla^{\prime }\cdot c-\frac{{\left(kH\right)}^{2}}{2Re}\left(\sum_{i=1}^{i=4}z_{i}{c^{\prime }}_{i}\right)\nabla^{\prime }\psi^{\prime }$$
(9)$$u^{\prime }\cdot \nabla^{\prime }c-\left(c\cdot \nabla^{\prime }{u^{\prime }}^{T}+\nabla^{\prime }u^{\prime }\cdot c\right)=-\frac{1}{Wi}\left(1+\varepsilon \left(tr\left(c\right)-3\right)\right)\left(c-I\right)$$
(10)$${\nabla^{\prime }}^{2}\psi^{\prime }=\frac{{\left(kH\right)}^{2}}{2}\left(z_{1}{c^{\prime }}_{1}+z_{2}{c^{\prime }}_{2}+z_{3}{c^{\prime }}_{3}{+z}_{4}{\;c\;\prime }_{4}\right)$$
(11)$$\nabla^{\prime }\cdot \left(u^{\prime }{c^{\prime }}_{i}-{D^{\prime }}_{i}\nabla^{\prime }{c^{\prime }}_{i}-z_{i}{D^{\prime }}_{i}{\nabla^{\prime }}_{i}\psi^{\prime }\right)=0,i=1..,4$$

In the above, $u^{\prime }$, $p^{\prime }$, $\psi^{\prime }$, and ${c^{\prime }}_{i}$ are dimensionless velocity, pressure, electric potential, and ion concentration, respectively. The Debye length is $k^{-1}=\sqrt{\varepsilon_{f}RT/\sum_{i=1}^{4}F^{2}z_{i}{}^{2}C_{0}}$. The viscosity ratio $\beta =\eta_{s}/\eta_{0}$ is the ratio of the solvent viscosity to the total viscosity. The dimensionless Reynolds number is $Re=2\rho u_{0}H/\eta_{0}$, and the Weissenberg number is $Wi=\lambda u_{0}/2H$.

The boundary conditions are given below.

(1) On the charged wall: the boundary is nonslip and ion-impenetrable and bears a surface charge density $\sigma_{0}$, and others are zero gradient.
(12)$$u^{\prime }=0,n\cdot \nabla^{\prime }\psi^{\prime }=\sigma_{0}\cdot \frac{HF}{RT},n\cdot \nabla^{\prime }{\;c\;\prime }_{i}=-z_{i}{c^{\prime }}_{i}n\cdot \nabla^{\prime }\psi^{\prime },n\cdot \nabla^{\prime }c=0$$

(2) At the anode (or cathode): the potential difference is *V*_0_, the pressure is 0, and each ionic concentration maintains its bulk value.
(13)$$n\cdot \nabla^{\prime }u^{\prime }=0,\psi^{\prime }=V_{0}\cdot \frac{F}{RT}(or0),p^{\prime }=0,{c^{\prime }}_{i}=1,n\cdot \nabla^{\prime }c=0$$

As the problem is symmetric, at the centerline of channel *x* = *H*, zero gradient is imposed on all variables.

## 3. Numerical Method and Code Validation

The high Weissenberg number problem (HWNP) loses numerical accuracy and stability under relatively high *Wi* [35,52]. The Log Conformation Reformulation (LCR) method [52,53] has proven to be one of the most effective strategies for overcoming this problem, and the procedure is presented as follows.

As the conformation tensor c is a symmetric positive definite (SPD) matrix, its matrix logarithm exists as
(14)$$\Psi =\log(c)=R^{T}\log(\Lambda )R$$
where Λ is a diagonal matrix whose diagonal elements are the eigenvalues of ***c***; and ***R*** is an orthogonal matrix composed of eigenvectors of ***c***.

Then, the evolution Equation (9) for the conformation tensor ***c*** can be reformulated in terms of this new variable Ψ as
(15)$$u^{\prime }\cdot \nabla^{\prime }\psi -\left(\Omega \cdot \Psi -\Psi \cdot \Omega \right)-2B=-\frac{1}{Wi}e^{-\Psi }\left(1+\varepsilon \left(tr\left(e^{\Psi }\right)-3\right)\right)\left(e^{\Psi }-I\right)$$
where Ω and B are the anti-symmetric matrix and the symmetric traceless matrix of the decomposition of the velocity gradient tensor $\nabla^{\prime }u^{\prime }$, as derived by Fattal and Kupferman [53] and Zhang et al. [54]. After Ψ is solved, the conformation tensor ***c*** can be recovered from the matrix-exponential of Ψ as
(16)c=exp(Ψ)

A new solver which solves the above equations (Equations (7)–(16)) was created based on the open source computational fluid dynamics (CFD) software OpenFOAM (version 6.0, The OpenFOAM Foundation Ltd, London, UK). Quick, Gauss Linear, and MINMOD schemes are used to discretize the convection terms in Equations (8), (11) and (15), respectively. The coupling of velocity and pressure fields is solved by splitting of operators (PISO) in Equation (8). Orthogonal mesh is used with much denser mesh distributed near the charged wall.

In addition, in order to verify the accuracy of the new solver for the viscoelastic EOF, we compare the numerical results with those of Afonso et al. [37], who simplified the analytical solution of EOF. The simplified PTT (sPTT) model in the two-dimensional microchannel assumes a low zeta potential and a thin EDL, so a simplified Poisson–Boltzmann equation can be used. The geometry of the channel is set to height *H* = 100 nm and length *L* = 300 nm. The solvent viscosity is set to zero and can be compared to the sPTT model in the reference. The other parameters are set as follows: *D*_1_ = 1.96 × 10^−9^ m^2^·s^−1^, *D*_2_ = 2.03 × 10^−9^ m^2^·s^−1^, *T* = 300 K, *F* = 96485 C·mol^−1^, *ε_f_* = 7.08 × 10^−10^ CV^−1^·m^−1^. The potential at the inlet is set to 0.05 V, and the outlet is grounded. The zeta potential on the wall is set to −4.36 mV. We define the dimensionless parameter $kH=\frac{H}{\lambda_{D}}=\sqrt{\frac{H^{2}\sum_{i=1}^{2}F^{2}z_{i}{}^{2}C_{0}}{\varepsilon_{f}RT}}$, which represents the ratio of the height of the channel to EDL thickness.

Figure 2 shows the predicted dimensionless x-component velocity distribution in the middle of the channel at *kH* = 15. The corresponding values are analytical solutions for Newtonian fluids (*Wi* = 0) and viscoelastic fluids at *ε* = 1 and various *Wi*. It can be seen that the velocity distribution is plug-like in *kH* = 15, and it increases with higher *Wi*. The numerical results are in good agreement with the analytical solutions of Afonso et al. [31] for Newtonian and viscoelastic fluids under different *Wi*.

## 4. Results and Discussion

The validated solver is then used to solve the (long enough) microchannel electrodynamic behavior of four ions for different *Wi* numbers, extensibility parameters, and viscosity ratios. The non-simplified PNP model is used to describe the ionic transport and potential distribution, and the LPTT constitutive model characterizes the fluid properties. The channel height is 2*H* = 30 μm and the length is *L* = 200 μm. A finer mesh is created around the charged wall to capture its EDL. Typically, the total number of elements is around 88,500 to achieve convergence and grid-independent results. Considering the fully developed EOF under different *Wi*, the diffusion rates of K^+^ and Cl^−^ ions are *D*_1_ = *D*_2_*=* 1 × 10^−9^ m^2^·s^−1^, at different KCl bulk concentrations, *C*_0_ = 0.01, 0.1 and 1 mM. We consider the near-neutral solution (pH = 7.5), H^+^ concentration *C*_3_ = 10^−pH + 3^ mM, OH^−^ concentration *C*_4_ = 10^−(14−pH) + 3^ mM, and the diffusion rates [25,26] are *D*_3_ = 9.31 × 10^−9^ m^2^·s^−1^, and *D*_4_ = 5.30 × 10^−9^ m^2^·s^−1^, surface charge density $\sigma_{0}=0{.03\;\mathrm{mC}/m}^{2}$, and the applied potential *V*_0_ = 1 V. Other parameters: extensibility parameter ε = 0.25, viscosity ratio *β* = 0.1. We maintain the above settings in the study unless otherwise stated.

### 4.1. The Influence of Wi Numbers

Figure 3 shows the dimensionless velocity considering *kH* = 15. It can be seen that the velocity increases sharply and reaches its magnitude near the charged wall, showing a plunger-like profile. The larger the *Wi* number, the more pronounced the plunger profile. This is because the thickness of EDL is much smaller than the height of the channel, so the charge is neutral outside the EDL region, and the velocity of the ion in the EDL is very slow when it is attracted by the Coulomb force. The maximum velocity for *Wi* = 3 at the centerline of viscoelastic fluid is 5 times that of Newtonian fluid. With the increase in *Wi*, the velocity increases monotonously in the whole range of *Wi*.

Figure 4 shows the dimensionless velocity considering *kH* = 250, with *Wi* values of 0, 1, 2, and 3, respectively. A sudden change was observed near the charge wall, and the plunger flow was more pronounced. The reason is that the thickness of the EDL is extremely small, the internal counter ions are tightly bound in the EDL, and the flow rate in the EDL is sufficiently negligible to be approximately zero. The amplitude increases with the increase in the Weissenberg number. The influence on the velocity is obvious with a low number, but the influence is gradually reduced with a higher *Wi*. As the *Wi* increases, the flow rate shows a monotonous increase over the entire *Wi* range studied. The small EDL thickness (*kH* = 250) is reflected in the suddenly changed *U*^*^. It can be seen that the influence of the *Wi* number on the flow rate is huge, and the viscoelastic fluid velocity (*Wi* = 3) is more than 9 times that of the Newtonian fluid.

Figure 5 shows the shear stress ($\tau_{xy}$) values under different *Wi*. It can be seen that the shear stress is zero at the center of the channel away from the charged wall and sharply increases in the near-wall EDL (y-axis from 0.86 to 1.0 y/H). Obviously, the shear stress is also related to the *Wi*. As the *Wi* number increases, the viscoelasticity of the fluid increases. When the *Wi* is 0, it approximates Newtonian fluid and exhibits the maximum near-wall stress. With the increase in the *Wi*, the flow velocity increases near the charged wall, the shear rate (velocity gradient) increases, and the local viscous force decreases. Therefore, the shearing effect is weakened near the charged wall with the increasing *Wi*. This is due to the behavior of shear-thinning fluids (LPTT).

### 4.2. Analysis of Different EDL Thicknesses

Figure 6 shows the dimensionless velocity of the near wall (0.13 H) at different *kH* with *Wi* = 1. As *kH* increases, the EDL thickness decreases monotonously. It can be seen that when the EDL thickness is gradually increased, a similar plunger-like velocity distribution gradually transitions to the parabolic velocity. At low concentrations, the EDL overlap trend is more obvious. Therefore, the velocity distribution shows a large unevenness toward the centerline. The flow rate is higher due to the increase in counter ions throughout the channel compared with the results for thin EDL thickness.

Figure 7 shows the near-wall concentrations of K^+^, H^+^, Cl^−^, and OH^-^ at different EDL thicknesses, dimensionless concentration *c*_i_* = *c*_i_/*c*_i0_. In general, the cations near the charged wall are greater than the bulk concentration, and the anions are less than the bulk concentration. With the increase in *kH* in the steady state, the cations are attracted to the charged wall, while the anions are repelled. The concentration of cations increases rapidly, nearly 1.5 times the bulk concentration near the wall surface, while the anions are repelled, decreasing along the center to the wall surface, until the near-wall concentration is close to 0.68 times the bulk concentration. In addition, the concentration (*kH* = 250) indicates that the EDL is extremely thin, and the stern layer is closely arranged.

Figure 8 shows the net charge density index $rhoD=|\frac{\rho_{e}}{\rho_{e}{}_{{}_{\max}}}|$ near the wall. When it is smaller, there are many regions with a charge density index >0.5, indicating that the anions and cations are relatively dispersed near the wall and ion aggregation is low. With the increase in *kH*, the region with charge density >0.5 decreases, indicating that the degree of aggregation of anions and cations is high, and most of the regions approximate their bulk concentration. When *kH* = 250, the charge density is very small, and the anions and cations are assembled in a layer of two to three ions. The trend is very consistent with the ion distribution characteristics shown in Figure 7.

Figure 9 is a plot of the net mobile ions ($Cnet=\sum_{i=1}^{4}z_{i}c_{i}$ ) trend in cross-section for various background salt concentrations (or EDL thicknesses). Since the wall is negatively charged, more cations are attracted to the wall due to static electricity, while the anions are repelled. This phenomenon increases as *kH* increases. In this case, as the concentration of the KCl solution increases, the ion density of the surface increases greatly, causing a more considerable amount of cations to collect near the wall, thereby repelling a large amount of co-ions (anions). When *kH* = 250, the ions drop abruptly and the cations gather densely near the wall surface, similar to that shown in Figure 8. The curve of *kH* = 50 and *kH* = 150 is between high and low EDL thicknesses, and the trend is in accordance with the concentration curve near the charged wall.

### 4.3. Rheological Parameter Effects of LPTT Model on Flow Velocity

The EOF of a viscoelastic fluid depends on the rheological parameters. Figure 10 shows the dimensionless velocity profiles for different viscosity ratios (*β*), with *Wi* = 1 and *kH* = 50. As the viscosity ratio increases, the magnitude of the velocity decreases correspondingly, and the velocity for *β* = 0.05 is about 3 times higher than that at *β* = 0.50. The flow mainly depends on the solvent viscosity, and the lower the solvent viscosity (proportion), the higher the velocity. This can be explained by the effect of macromolecular agglomeration on viscoelastic fluids.

Figure 11 shows the shear stress values of the xy plane under different viscosity ratios (*β*). It can be seen that the shear stress is almost zero in the central area of the entire channel and increases sharply in the vicinity of the charged wall. The numerical results show that as the viscosity ratio (*β*) increases, the amplitude of the shear stress decreases. This is due to the shear thinning characteristics of the LPTT fluid, which affect the increase in the shear rate at the wall surface, resulting in an increase in velocity. As the viscosity ratio decreases, the shear stress $\tau_{xy}=\eta_{0}\frac{\partial u}{\partial y}$ near the wall increases due to the increase in the ratio of solute viscosity, high gradients of velocity, molecular agglomeration, and macromolecular elasticity.

Figure 12 shows the cross-section EOF under different extensibility parameters (*ε*). The velocity at *ε* = 1 is twice the value at *ε* = 0.05. The smaller extensibility parameters, the greater the contribution to the velocity. However, with the increase in the stretching parameters, the contribution is gradually weakened.

## 5. Conclusions

The EOF of viscoelastic fluids in long microchannels was numerically studied, and the effect of the rheological properties of LPTT fluids was considered on fully developed EOF with near-neutral condition (pH = 7.5 and four ions). The nonlinear PNP equation is used to describe the potential and ion concentration distribution without using the simplified PB or DH model assumptions. The overall conclusions are as follows:

(1) When the EDLs do not overlap, the velocity distributions of different Weissenberg numbers are all plunger-like, and the influence of the *Wi* number on the flow velocity is significant. The velocity of the viscoelastic fluid was observed to increase significantly compared with Newtonian fluid. For thick electric double layers (*kH* = 15), the maximum velocity at *Wi* = 3 at the centerline of the viscoelastic fluid is 5 times that of Newtonian fluid. As the *Wi* increased, the flow rate showed a monotonous increase over the entire *Wi* range studied. For the thin double layer (*kH* = 250), the viscoelastic fluid with a *Wi* value of 3 has a maximum velocity that is 9 times that of Newtonian fluid.

(2) Newtonian fluid (*Wi* = 0) has the largest shear-stress at the surface charged wall, and as the *Wi* number increases, the local viscous force decreases, which weakens the shear-thinning behavior. As the velocity increases with the increase in *kH*, the concentration of cations is close to 1.5 times that of the bulk concentration, and the concentration of anions is close to 0.68 times that of the bulk concentration. When the *kH* = 250, the anions and cations are concentrated in a layer of two to three ions.

(3) As the viscosity ratio (*β*) increases, the magnitude of the dimensionless *U*^*^ decreases correspondingly, and the velocity at *β* = 0.05 is about 3 times that at *β* = 0.50. With the viscosity ratio dropping, the shear-stress near the wall increases. This is due to the large solute viscosity, the effects of molecular agglomeration and macromolecular elasticity. The velocity at *ε* = 1 is twice that at ε = 0.05. When the extensibility parameter is large, the contribution is gradually weakened.

The research results may contribute to experimental research on the latest PoCT devices. These devices can be used in biological fluid detection and new generation microchip applications, from the development of medical devices with artificial flow driving to portable diagnostic kits.

## Figures and Tables

**Figure 1 micromachines-10-00881-f001:**
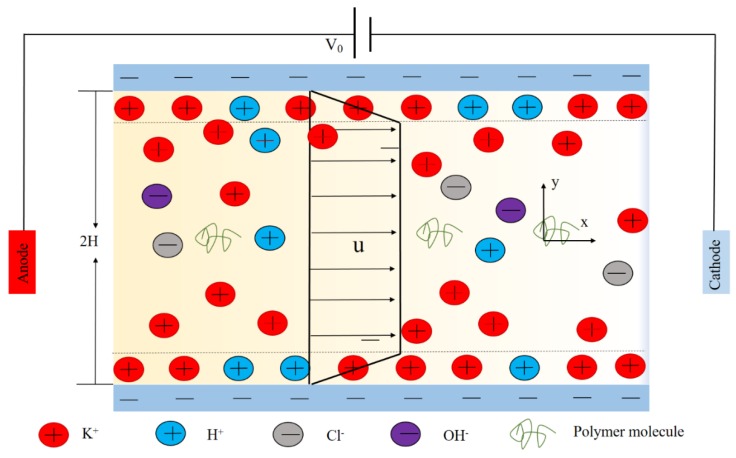
Schematic diagram of viscoelastic fluid electroosmotic flow (EOF).

**Figure 2 micromachines-10-00881-f002:**
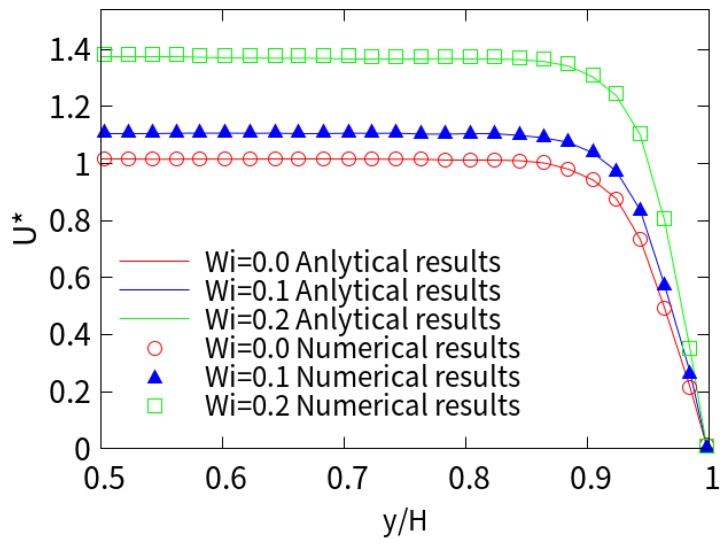
Comparison between the numerical (symbol) and analytical (solid line) results of the axial EOF velocity at different Weissenberg numbers (*Wi*).

**Figure 3 micromachines-10-00881-f003:**
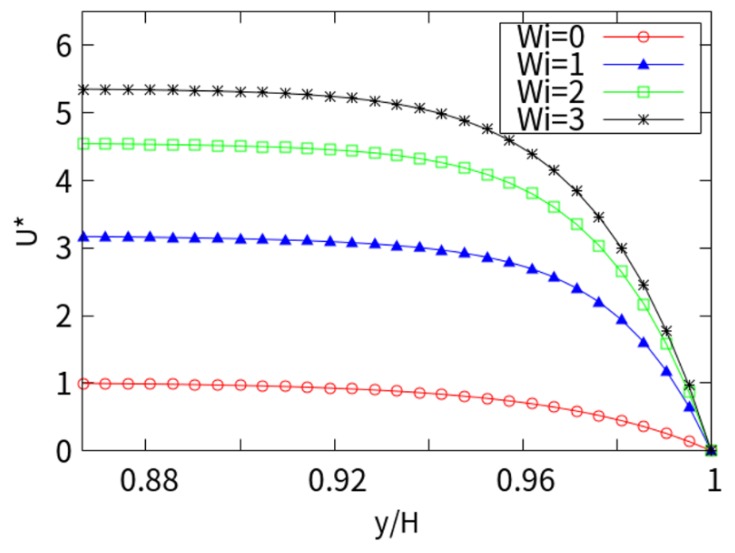
Cross-section velocity in the x direction with *kH* = 15.

**Figure 4 micromachines-10-00881-f004:**
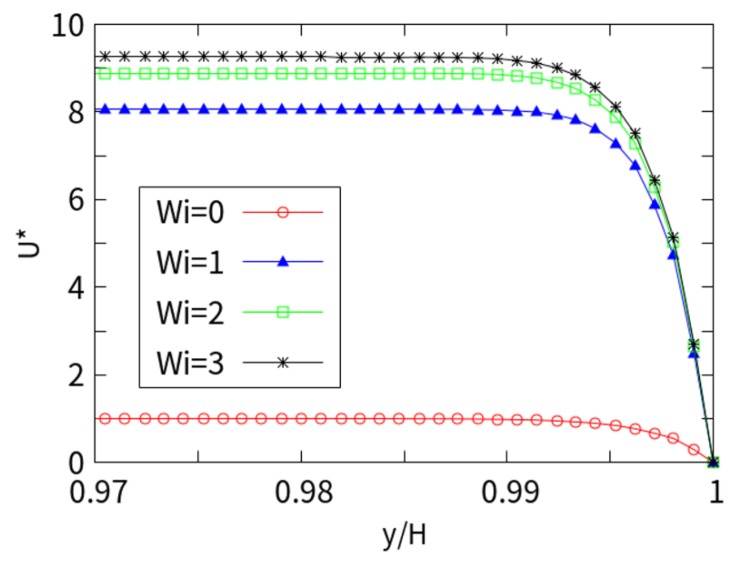
Cross-section velocity in the x direction with *kH* = 250.

**Figure 5 micromachines-10-00881-f005:**
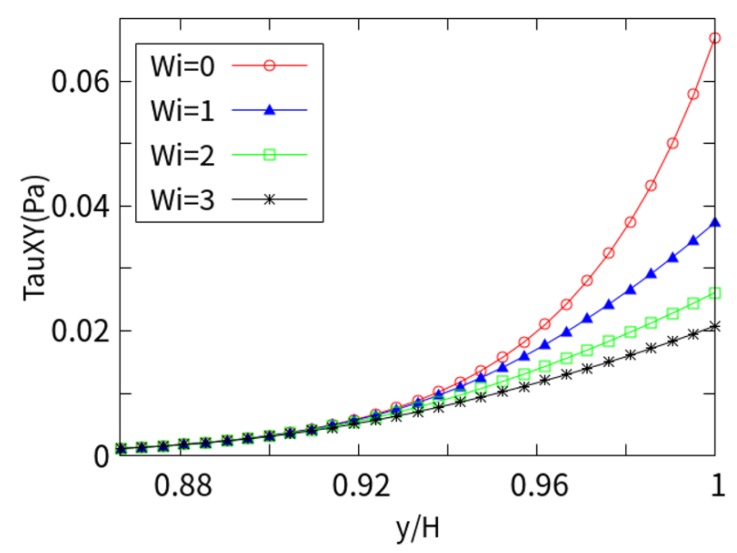
Shear stress at different *Wi* numbers with *kH* = 15.

**Figure 6 micromachines-10-00881-f006:**
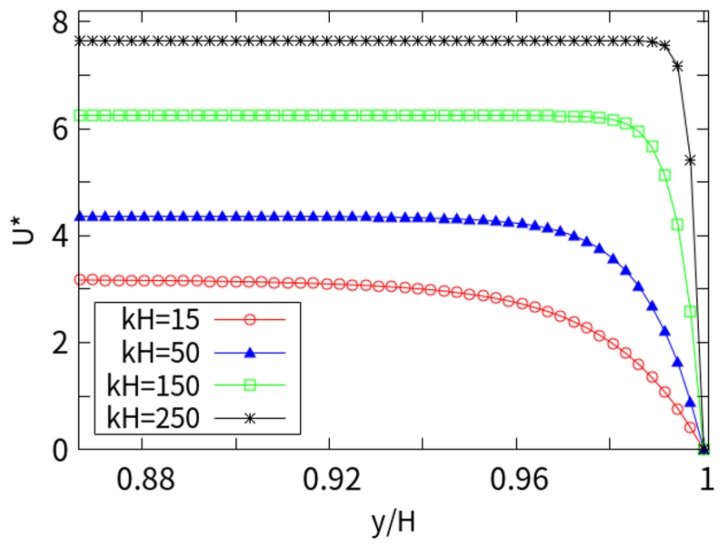
Velocity distribution of different electrical double layer (EDL) thicknesses.

**Figure 7 micromachines-10-00881-f007:**
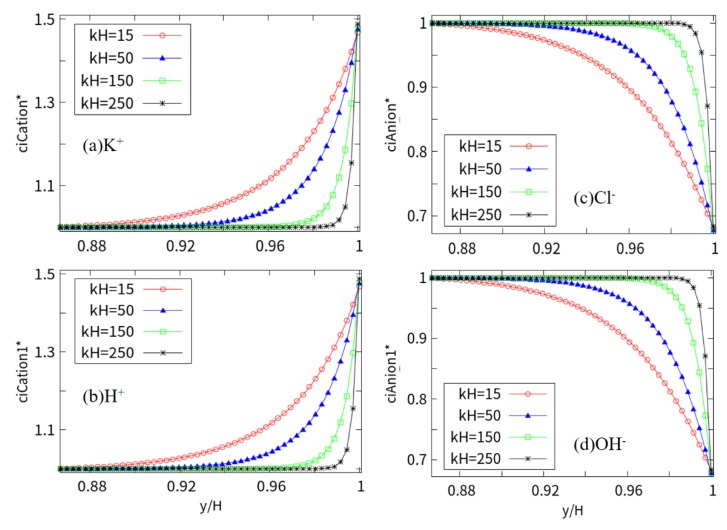
Wall distribution of anions and cations at different *kH* (the vertical axis indicates the dimensionless concentration *c*_i_* = *c*_i_/*c*_i0_).

**Figure 8 micromachines-10-00881-f008:**
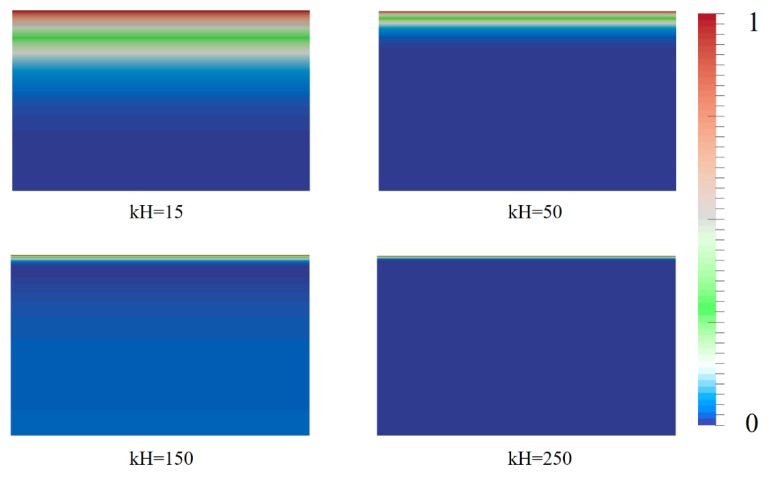
Net charge density index at different *kH*.

**Figure 9 micromachines-10-00881-f009:**
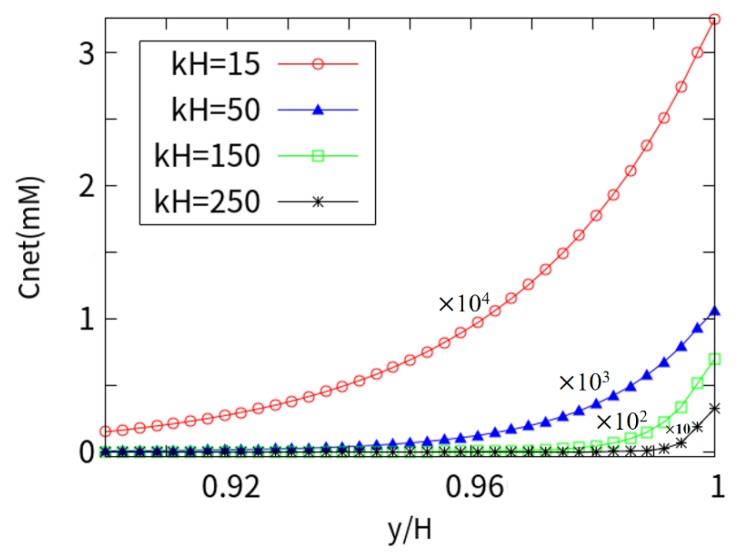
Net mobile ions trend in cross-section.

**Figure 10 micromachines-10-00881-f010:**
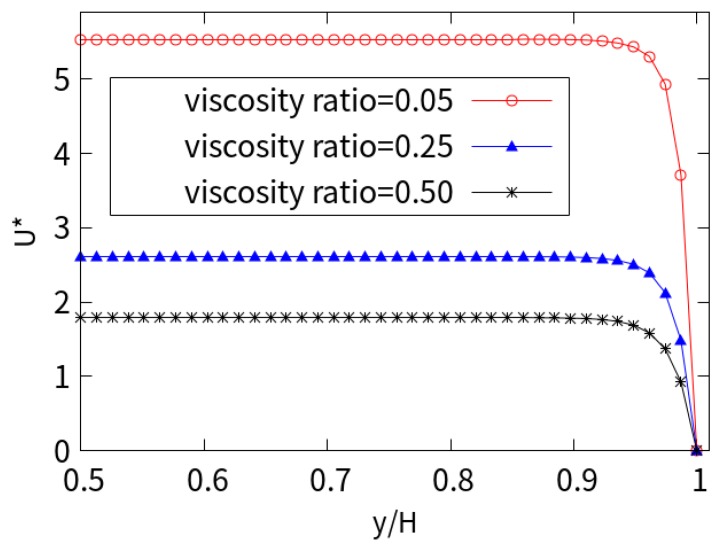
Near-wall velocity for different viscosity ratios.

**Figure 11 micromachines-10-00881-f011:**
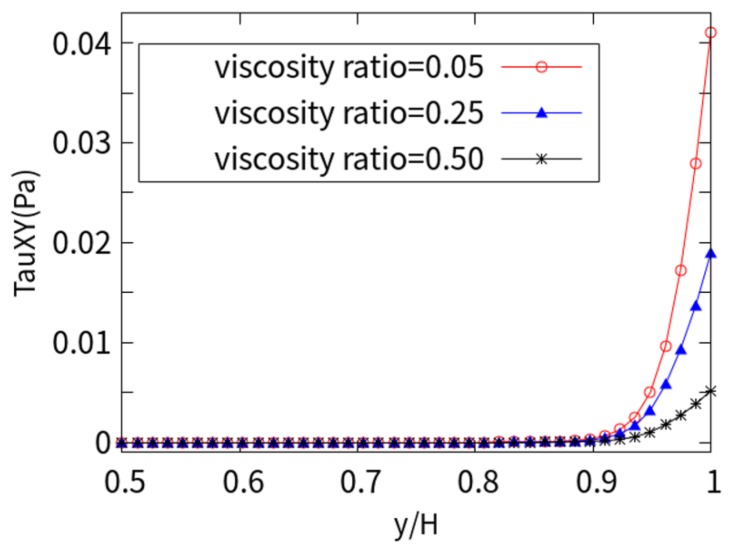
Shear stress for different viscosity ratios.

**Figure 12 micromachines-10-00881-f012:**
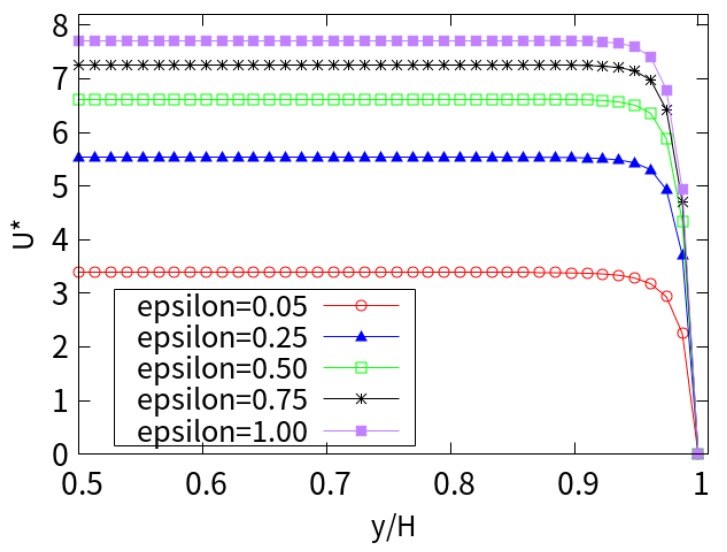
Near-wall velocity for different extensibility parameters.

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
