# Peer review of "Electroosmotic Flow Behavior of Viscoelastic LPTT Fluid in a Microchannel"

_micromachines, 2019, doi:10.3390/mi10120881_

Round 1
Reviewer 1 Report
The manuscript presents numerical results of electro-osmotic flow of viscoelastic fluids described by the LPTT model.
It is not clear exactly what is the novelty of this work. The simulations are only for simple cases, so they are not very demanding in terms of computational resources.
The authors state that they are "using a new solver implemented in an open source CFD software OpenFOAM", but no description of the novelty is presented. The governing equations are presented in detail, but the numerical implementation is not described. There is no added value for the reader besides knowing that the results agree with other works. The usefulness of this work would be from the description of the new numerical method.
Was the numerical model based on the viscoelasticFluidFoam implementation in openFOAM, or any other?
Unless the authors present the numerical method in detail, and describe the novelty of such method, I do not see significant value in the manuscript to be published in Micromachines.
Reviewer 2 Report
Please see my comments in the appended document.
I have some questions that I would like to be adressed before publication.

Reviewer 3 Report
Chen et al. numerically investigated the EOF behavior of viscoelastic fluids since most polymer solution and biological fluid are not a Newtonian fluid. The behavior has been expressed as a function of different fluid behavior index, Weissenberg number, EDL thickness, species, viscosity ratio, extensibility parameter, and etc.
There are some questions that future readers may have:
In line 110, there is no Nj in equation (6). In line 112 and 123, are u and Wi vectors? In line 146, what does large or small fluid behavior index mean? I think its explanation can be found in line 153 & 154. In line 145 and 177, are kH =15 and kH=250 typical number? For what conditions and/or fluids do you get those numbers? Similarly, in line 173, Wi = 0 seems to be Newtonian fluid while Wi = 1, 2, 3 seem to be non-Newtonian fluid. What would be the big difference between Wi =1 and Wi =3? In Figure 12, what kind of fluid does different epsilon represent?Author Response
Please see the attachment.

Round 2
Reviewer 2 Report
The manuscript was improved as suggested. The author were careful while performing the corrections.
I therefore recommend its publication.